# On Sparsity and Sub-Gaussianity in the Johnson-Lindenstrauss Lemma

**Aurélien Garivier**  *aurelien.garivier@ens-lyon.fr*
*ENS de Lyon UMPA UMR 5669 /*
*LIP UMR 5668 46 allée d'Italie*
*F-69364 Lyon cedex 07*
 *and*
**Emmanuel Pilliat**  *emmanuel.pilliat@ensai.fr*
*ENSAI Rennes*

**Reviewed on OpenReview:** *https: // openreview. net/ forum? id= Znaty8V3a3&noteId= rFSbwoWelg*

## Abstract

We provide a simple proof of the Johnson-Lindenstrauss lemma for sub-Gaussian variables. We extend the analysis to identify how sparse projections can be, and what the cost of sparsity is on the target dimension. The Johnson-Lindenstrauss lemma is the theoretical core of the dimensionality reduction methods based on random projections. While its original formulation involves matrices with Gaussian entries, the computational cost of random projections can be drastically reduced by the use of simpler variables, especially if they vanish with a high probability. In this paper, we propose a simple and elementary analysis of random projections under classical assumptions that emphasizes the key role of sub-Gaussianity. Furthermore, we show how to extend it to sparse projections, emphasizing the limits induced by the sparsity of the data itself.

## 1 Introduction

The celebrated Johnson-Lindenstrauss lemma Johnson & Lindenstrauss (1984) ensures the existence low-distortion embeddings of points from high-dimensional into low-dimensional Euclidean space. If $x_1, \ldots, x_n \in \mathbb{R}^p$, where $p$ is a (large) integer, and if $\epsilon > 0$ is a tolerance parameter, then there exists a matrix $A$ in the set $\mathcal{M}_{d,p}(\mathbb{R})$ of real matrices with $d$ rows and $p$ columns such that

$$\forall 1 \leq i, j \leq n, \quad (1 - \epsilon)\|Ax_i - Ax_j\|^2 \leq \|x_i - x_j\|^2 \leq (1 + \epsilon)\|Ax_i - Ax_j\|^2 \tag{1}$$

as soon as

$$d \geq \frac{8 \ln(n)}{\epsilon^2 - \epsilon^3} . \tag{2}$$

The classical proof of this result is an elegant illustration of the Probabilistic Method Alon & Spencer (2004): when drawing the entries of $A$ at random from independent Gaussian distributions, Property (11) is satisfied with positive probability as soon as the output space is large enough. It results from a simple deviation bound for the chi-square distribution, and hence builds on the specificity of the Gaussian distribution. This proof is not only mathematically remarkable, but it also gives mathematical foundations for *random projections*, a simple and computationally efficient dimensionality reduction technique in unsupervised machine learning (see e.g. Bingham & Mannila (2001); Indyk (2001); Vempala (2004); Sarlos (2006); Clarkson & Woodruff (2009) and references therein).

In 2001, Achlioptas (2001) showed that random projections can easily be extended to non-Gaussian matrices. In particular, Rademacher, or $\{-1, 0, 1\}$-valued entries can just as well be chosen, leading to even simpler algorithms suitable for database applications. The proof provided in Achlioptas (2001) relies on

moment bounds and is somewhat specific to those two families of distributions. It was improved in Matoušek in Matoušek (2008) with more general arguments. It is generally considered Li et al. (2006) that "a uniform distribution is easier to generate than normals, but the analysis is more difficult". Even faster methods for sparse data on streams were then devised Ailon & Chazelle (2009); Charikar et al. (2004); Kane & Nelson (2014) using random hashing constructions and more involved moment bounds. Very recently and concurrently to our work, Li (2024) has proposed a unified analysis of sparse Johnson-Lindenstrauss methods based on the Hanson-Wright inequality, while Høgsgaard et al. (2024) tries to identify the optimal sparsity rate in the data as a function of the dimension $d$, the number of points $n$ and the tolerance parameter $\epsilon$.

Our goal is to make the analysis of sparse random projections accessible to a wider audience, and the main contribution of this paper is two-fold. The first purpose is to highlight that *sub-Gaussianity* is in fact an elementary property of random matrix entries that suffices to ensure the success of random projections. The possibility of using sub-Gaussian entries was (to the best of our knowledge) first mentioned in Matoušek (2008). Unlike Li (2024), our analysis is entirely elementary: exploiting sub-Gaussianity in an original way, it does not require the Hanson-Wright inequality and relies solely on Chernoff and Bernstein bounds. Compared to Matoušek (2008); Chafai et al. (2012), the central argument is simplified, the approach is new, and we pay special attention to the constant in the exponential deviation bound. The folk theorem (stated for example in Boucheron et al. (2013)) is that using sub-Gaussian entries instead of normal laws is possible at the price of "just a multiplicative constant" in the target dimension. Focusing on this multiplicative constant, we show what distributions can be used to obtain *exactly the same guarantees* as with Gaussian entries. Trying to provide an optimal compromise between tightness and clarity, we also give intuitions and intermediate products that may be overseen with the general point of view of Orlicz norms. We give a simple proof that any 1-sub-Gaussian law with variance 1 offers the same guarantees as the Gaussian law. We provide a simple and critical analysis for Achlioptas' limit of $1/3$ of non-zero coefficients, and under what conditions it is possible to go further in the understanding of much sparser random projections. Interestingly, our treatment of the lower and upper bound of 11 is not totally symmetric. While the upper deviations of sub-Gaussian variables can be handled by Chernoff's bound just as those of the Gaussian law, the lower deviations can obviously be much smaller (after all, constant variables are sub-Gaussian) and hence require a different argument. To highlight the connection to the Hanson-Wright inequality, we propose in the final section an elementary proof of this inequality for sub-Gaussian vectors when the quadratic form is positive definite. It relies on the same tools as the other proofs of this paper, and it can be read independently.

The second purpose of this paper is to build on this analysis to clearly emphasize the conditions on the data under which much sparser random projection matrices can be considered. Our approach provides a clear decomposition of noise into a sparse component $\zeta$ and a sub-Gaussian component $U$. The (pretty intuitive) take-home message is that the distances are preserved if and only if the proportion of non-zero entries in the projection matrix $A$ multiplied by the number of significant coefficients in each vector $x_i$ is sufficiently large: we propose a quantitative and mathematically rigorous formulation of this intuition. An analysis of the same flavor was established by Matoušek (2008) for a specific distribution on the sparse projection $A$, in the case $n = 2$. While the arguments in Matoušek (2008) could be generalized to larger values of $n$ and other distributions, we present a new perspective through detailed proofs in the sparse and dense cases, with careful attention to constants in the dense case. Furthermore, we fully elaborate on the lower bound argument outlined in that article. Specifically, we establish that the condition on the proportion of non-zero coefficients of $A$ in our main result is essentially not improvable.

The paper is organized as follows. The pedagogical Section 2 includes our analysis of random projections without assumptions on the data: Section 2.1 includes a deviation bound for the averages of squared sub-Gaussian variables which is applied Section 2.2 to derive the classical Johnson-Lindenstrauss lemma for sub-Gaussian random matrices. We discuss in Section 2.3 a few examples of choices of the distribution $P$ for random projections. Section 3 investigates the possibility of much sparser projection matrices and of the theoretical limit to the minimal sparsity. Theorem 2, with its rather simple proof in Section A, extends the previous analysis with minimal changes to sparse matrices. Theorem 3 gives the order of magnitude of the minimal allowed sparsity to obtain a quasi-isometry with high probability, at the price of poly-logarithmic terms. The optimality of this result is discussed in Section 3.3. A connection to the Hanson-Wright inequality is proposed in Section 4, before the full proofs of the main theorems in the appendix (Section A).

## 2 Data-agnostic random projections

We recall in this section known but fundamental results that are of constant use in the sequel. The originality lies in the fact that the Johnson-Lindenstrauss lemma is stated from the start for sub-Gaussian variables. Furthermore, we were not able to find anywhere else the elegant derivation of 3 written like this. Section 2.3 contains a simpler derivation of results published in Achlioptas (2001), with a discussion on their optimality.

### 2.1 Chernoff's method for squared sub-Gaussian variables

Let $X$ be a random variable assumed to be 1-sub-Gaussian, which means that $\forall \lambda \in \mathbb{R}, \mathbb{E}\left[e^{\lambda X}\right] \leq e^{\lambda^2/2}$. This implies in particular that $\mathbb{E}[X] = 0$ and that $\mathbb{V}\mathrm{ar}[X] \leq 1$. We derive in this section a deviation bound for the empirical mean of independent copies of $X^2$:

**Proposition 1** *If $X_1, \ldots, X_n$ are iid 1-sub-Gaussian random variables with variance 1, then*

$$\mathbb{P}\left(1 - \epsilon \leq \frac{X_1^2 + \cdots + X_d^2}{d} \leq 1 + \epsilon\right) \leq 2e^{-d\left(\frac{\epsilon^2 - \epsilon^3}{4}\right)} .$$

For Gaussian variables, this is a well-known application of Chernoff's method that is to be found in many probability textbooks. Inspired in particular by Theorem 2.6 of Wainwright (2019), we propose an extension to sub-Gaussian variable with an argument that is (as far as we know) original. The proof requires to treat the upper- and the lower bound separately, which is done in the two following subsections.

### 2.1.1 Proof of the upper bound

Chernoff's method requires to bound the exponential moments $\mathbb{E}\left[e^{\ell X^2}\right]$ of $X^2$ with $\ell > 0$ for the right deviations and with $\ell < 0$ for the left deviations. We start with the right deviations, for which we will see right away that a reduction to the Gaussian case is possible without further assumption. Following Wainwright (2019) (Theorem 2.6), and remarking that for all $x \in \mathbb{R}$, and $\ell > 0$,

$$e^{\ell x^2} = \int_{-\infty}^{\infty} e^{\lambda x} \frac{e^{-\frac{\lambda^2}{4\ell}}}{2\sqrt{\pi\ell}} \, d\lambda \, ,$$

if $X$ is 1-sub-Gaussian we obtain by Fubini's theorem that for every $\ell \in (0, 1/2)$

$$\mathbb{E}\left[e^{\ell X^2}\right] = \mathbb{E}\left[\int_{-\infty}^{\infty} e^{\lambda X} \frac{e^{-\frac{\lambda^2}{4\ell}}}{2\sqrt{\pi\ell}} \, d\lambda\right] = \int_{-\infty}^{\infty} \mathbb{E}\left[e^{\lambda X}\right] \frac{e^{-\frac{\lambda^2}{4\ell}}}{2\sqrt{\pi\ell}} \, d\lambda$$

$$\leq \int_{-\infty}^{\infty} e^{\frac{\lambda^2}{2}} \frac{e^{-\frac{\lambda^2}{4\ell}}}{2\sqrt{\pi\ell}} \, d\lambda = \int_{-\infty}^{\infty} e^{-\frac{\lambda^2(1-2\ell)}{4\ell}} \frac{d\lambda}{2\sqrt{\pi\ell}} = \frac{1}{\sqrt{1-2\ell}} \, ,$$

which holds with equality if and only if $X \sim \mathcal{N}(0, 1)$. A slightly more elegant approach to establishing the above equality is to observe that for $G \sim \mathcal{N}(0, 1)$, Fubini's theorem implies that

$$\mathbb{E}_X\left[e^{\ell X^2}\right] = \mathbb{E}_X\left[\mathbb{E}_G\left[e^{\sqrt{2\ell}X\,G}\right]\right] = \mathbb{E}_G\left[\mathbb{E}_X\left[e^{\sqrt{2\ell}G\,X}\right]\right]$$

$$\leq \mathbb{E}_G\left[e^{\ell G^2}\right] = \frac{1}{\sqrt{2\pi}} \int_{\mathbb{R}} e^{\ell u^2} e^{-\frac{u^2}{2}} \, du = \frac{1}{\sqrt{1-2\ell}} \tag{3}$$

with equality if and only if $X \sim \mathcal{N}(0, 1)$.

Hence, all sub-Gaussian variables have exponential moments bounded by those of a Gaussian law, which permits the right-deviations to be handled the usual way. If $Z_1, \ldots, Z_d$ are independent random variables with the same distribution as $X^2$, then for every positive $\epsilon$, Markov's inequality implies that

$$\mathbb{P}\left(\frac{Z_1 + \cdots + Z_d}{d} \geq 1 + \epsilon\right) = \mathbb{P}\left(e^{\ell\left(Z_1 + \cdots + Z_d\right)} \geq e^{d\ell(1+\epsilon)}\right) \leq \frac{\mathbb{E}[e^{\ell Z_1}]^d}{e^{d\ell(1+\epsilon)}} = e^{-d\left(\ell(1+\epsilon) - \ln\mathbb{E}\left[e^{\ell Z_1}\right]\right)} \, .$$

The concave function $\ell \mapsto \ell(1 + \epsilon) + \frac{1}{2}\ln(1 - 2\ell) \leq \ell(1 + \epsilon) - \ln\mathbb{E}\big[e^{\ell Z_1}\big]$ is maximized at $\ell^*$ such that $1 + \epsilon = \frac{1}{1-2\ell^*}$, that is at $\ell^* = \frac{1}{2}\left(1 - \frac{1}{1+\epsilon}\right) = \frac{\epsilon}{2(1+\epsilon)}$. Hence, $\mathbb{P}\big(Z_1 + \cdots + Z_d \geq (1 + \epsilon)d\big) \leq e^{-d\,I(\epsilon)}$ with

$$I(\epsilon) = \ell^*(1 + \epsilon) - \ln\mathbb{E}\big[e^{\ell^* Z_1}\big] \geq \frac{\epsilon - \ln(1 + \epsilon)}{2} .$$

This expression can be slightly simplified in many different ways. Let us illustrate the very useful "Pollard trick": taking $g(\epsilon) = \epsilon - \ln(1 + \epsilon)$, since $g(0) = g'(0) = 0$ and since $g''(\epsilon) = 1/(1 + \epsilon)^2$ is convex, by Jensen's inequality

$$\frac{\epsilon - \ln(1 + \epsilon)}{\epsilon^2/2} = \int_0^1 g''(s\epsilon)2(1 - s)ds \geq g''\left(\epsilon\int_0^1 s\,2(1 - s)ds\right) = g''\left(\frac{\epsilon}{3}\right) ,$$

and hence $I(\epsilon) \geq \dfrac{\epsilon - \ln(1 + \epsilon)}{2} \geq \dfrac{\epsilon^2}{4\left(1 + \frac{\epsilon}{3}\right)^2} \geq \dfrac{\epsilon^2 - \epsilon^3}{4}$. In summary,

$$\mathbb{P}\left(\frac{Z_1 + \cdots + Z_d}{d} \geq 1 + \epsilon\right) \leq e^{-d\left(\frac{\epsilon^2 - \epsilon^3}{4}\right)} . \tag{4}$$

### 2.1.2 Proof of the lower bound

There is no hope to prove that $\mathbb{E}\left[e^{-\ell X^2}\right] \leq \frac{1}{\sqrt{1+2\ell}}$ for any $\ell > 0$ for all 1-sub-Gaussian distributions, since it is for example not the case if $X = 0$ almost surely. In the context of the Johnson-Lindenstrauss lemma, it is very natural to assume that the entries of the random matrix have variance 1, so that at least $\mathbb{E}\big[\|Ax_i - Ax_j\|^2\big] = \|x_i - x_j\|^2$. Under this assumption, it is maybe possible to bound the negative exponential moments bounded by those of the standard Gaussian and to conclude (as in the Gaussian case) by remarking that $I(-\epsilon) \geq I(\epsilon)$, i.e. that the left-deviations of the Chi-square are lighter than the right deviations. But we do unfortunately not have a proof for that.

Instead, we remark that if $\mathbb{Var}[X] = 1$, the sub-Gaussianity inequality $\mathbb{E}\big[e^{\lambda X}\big] \leq e^{\lambda^2/2}$ implies by Taylor expansion around $\lambda = 0$ that $\mathbb{E}[X^4] \leq 3$. Using that $e^{-u} \leq 1 - u + \dfrac{u^2}{2}$, we obtain that $\mathbb{E}\left[e^{-\ell X^2}\right] \leq 1 - \ell + \frac{3\ell^2}{2}$ and hence

$$\mathbb{P}\left(\frac{Z_1 + \cdots + Z_d}{d} \leq 1 - \epsilon\right) \leq e^{-d\left(\ell(-1+\epsilon) - \ln\left(1 - \ell + \frac{3\ell^2}{2}\right)\right)} .$$

Since $-\ln(1 - u) \geq u + u^2/2$,

$$\ell(-1 + \epsilon) - \ln\left(1 - \ell + \frac{3\ell^2}{2}\right) \geq \ell\epsilon - \frac{3\ell^2}{2} + \frac{(\ell - 3\ell^2/2)^2}{2} \geq \ell\epsilon - \ell^2 - \frac{3\ell^3}{2} = \frac{\epsilon^2}{4} - \frac{3\epsilon^3}{16}$$

for $\ell = \epsilon/2$. It follows that

$$\mathbb{P}\left(\frac{Z_1 + \cdots + Z_d}{d} \leq 1 - \epsilon\right) \leq e^{-d\left(\frac{\epsilon^2}{4} - \frac{3\epsilon^3}{16}\right)} \leq e^{-d\left(\frac{\epsilon^2 - \epsilon^3}{4}\right)} . \tag{5}$$

### 2.2 Application to the Johnson-Lindenstrauss lemma

Now that we have proved that squares of sub-Gaussian variables concentrate as well as squares of Gaussian variables, we recall for the sake of self-containment the argument that permits to obtain the Johnson-Lindenstrauss lemma with no assumption on the data:

**Theorem 1 (Johnson-Lindenstrauss Lemma)** *Let $x_1, \ldots, x_n \in \mathbb{R}^p$ and $\epsilon > 0$. For every $d \geq \frac{8\ln(n)}{\epsilon^2 - \epsilon^3}$ there exists a matrix $A \in \mathcal{M}_{d,p}(\mathbb{R})$ such that*

$$\forall 1 \leq i, j \leq n, \quad (1 - \epsilon)\|Ax_i - Ax_j\|^2 \leq \|x_i - x_j\|^2 \leq (1 + \epsilon)\|Ax_i - Ax_j\|^2 . \tag{6}$$

**Proof:** In the sequel, we assume that $A_{i,j} = T_{i,j}/\sqrt{d}$, $1 \leq i \leq d, 1 \leq j \leq p$, where the $(T_{i,j})$ are centered, standard independent variables of a 1-sub-Gaussian distribution $P$:

$$\mathbb{E}[T_{i,j}] = 0, \quad \mathbb{V}\mathrm{ar}[T_{i,j}] = 1, \quad \mathbb{E}\left[e^{\lambda T_{i,j}}\right] \leq e^{\frac{\lambda^2}{2}} .$$

For a vector $y \in \mathbb{R}^p$ of norm $\|y\|_2 = 1$, define $Y = Ay$ and for all $i \in \{1, \ldots, d\}$

$$Z_i = \sqrt{d}\, Y_i = \sum_{j=1}^{p} y_j T_{i,j} .$$

Then, as for all $\lambda \in \mathbb{R}$

$$\mathbb{E}\left[e^{\lambda Z_i}\right] = \prod_{j=1}^{p} \mathbb{E}\left[e^{\lambda y_j T_{i,j}}\right] \leq \prod_{j=1}^{p} e^{\frac{y_j^2 \lambda^2}{2}} = e^{\frac{\lambda^2}{2}} ,$$

$Z_i$ is 1-sub-Gaussian. Since

$$\|Ay\|^2 = \frac{1}{d} \sum_{i=1}^{d} \left(\sqrt{d} Y_i\right)^2 = \frac{1}{d} \sum_{i=1}^{d} Z_i^2 ,$$

Equations 4 and 5 yield:

$$\mathbb{P}\left(\|Ay\|^2 \notin [1-\epsilon, 1+\epsilon]\right) = \mathbb{P}\left(\frac{1}{d} \sum_{i=1}^{d} Z_i^2 > 1+\epsilon\right) + \mathbb{P}\left(\frac{1}{d} \sum_{i=1}^{d} Z_i^2 < 1-\epsilon\right) \leq 2\, e^{-d\left(\frac{\epsilon^2 - \epsilon^3}{4}\right)} \leq \frac{2}{n^2}$$

as soon as $d \geq \frac{8 \ln(n)}{\epsilon^2 - \epsilon^3}$. By the union bound and the above inequality applied to $y = \frac{x_i - x_j}{\|x_i - x_j\|}$ for all $i < j$ such that $x_i \neq x_j$,

$$\mathbb{P}\left(\bigcup_{1 \leq i < j \leq n} \left\{\left\|A(x_i - x_j)\right\|^2 \notin \left[(1-\epsilon)\|x_i - x_j\|^2, (1+\epsilon)\|x_i - x_j\|^2\right]\right\}\right) \leq \frac{n(n-1)}{n^2} < 1 ,$$

hence giving the desired conclusion. $\qquad\square$

Observe that the constant 8 in Condition 2 is the best that can be obtained from this proof. The dependency in $1/\epsilon^2$ also appears to be necessary, but the second-order term $\epsilon^3$ is slightly improvable. In the Gaussian case, the proof above allows to use

$$d = \frac{4 \ln(n)}{\epsilon - \ln(1+\epsilon)} \leq \frac{8 \ln(n)}{\epsilon^2} \left(1 + \frac{\epsilon}{3}\right)^2 ,$$

as we saw in Section 2.1.1. For sub-Gaussian variables, the simple expression 2 covers at the same time left- and right-deviations. Also observe that choosing $d \geq \frac{4 \ln(n^2/\delta)}{\epsilon^2 - \epsilon^3}$ permits Property 11 to hold with probability at least $1 - \delta$.

## 2.3 What distribution should we use in random projections?

We have seen that any 1-sub-Gaussian distribution of variance 1 presents just the same guarantees as the standard Gaussian for random projections. This is for example the case of $P = \frac{\delta_{-1} + \delta_1}{2}$, or of $P = \mathcal{U}\left(\left[-\sqrt{3}, \sqrt{3}\right]\right)$, which are very simple laws that are fast to sample from. Indeed, their exponential moment functions are $\cosh(\lambda)$ and $\sinh(\sqrt{3}\lambda)/(\sqrt{3}\lambda)$ respectively, which are upper-bounded by $e^{\lambda^2/2}$. One may wonder, after Achlioptas in Achlioptas (2001), how *sparse* a random projection matrix can be (sparse matrices require fewer computations).

**Proposition 2** *If $X$ is a 1-sub-Gaussian random variable of variance 1, then $P(X = 0) \leq 2/3$, with equality if and only $\mathbb{P}(X = -\sqrt{3}) = \mathbb{P}(X = \sqrt{3}) = \mathbb{P}(X \neq 0)/2$.*

**Proof:** Let us write $X = \zeta U$, where $\zeta \sim \text{Bern}(q)$ and $U$ is an independent centered random variable. The requirement $\mathbb{V}\text{ar}[X] = 1$ implies that $\mathbb{E}[U^2] = 1/q$. If $X$ is 1-sub-Gaussian, then $\mathbb{E}[X^4] = q\mathbb{E}[U^4] \leq 3$, and since $\mathbb{E}[U^4] \geq \mathbb{E}[U^2]^2 = 1/q^2$ this implies that $q \geq 1/3$. Moreover, the choice $q = 1/3$ is possible only if $\mathbb{E}[U^4] = \mathbb{E}[U^2]^2$, that is if $U^2 = 1/q$ almost surely. The choice

$$P = \frac{q}{2}\delta_{-\frac{1}{\sqrt{q}}} + (1-q)\delta_0 + \frac{q}{2}\delta_{\frac{1}{\sqrt{q}}} \tag{7}$$

with $q = 1/3$ is indeed the suggestion of Achlioptas, and it is 1-sub-Gaussian. The justification of this choice in Achlioptas (2001) involves pretty involved moment computations, while we here only need to check that for all $\lambda \in \mathbb{R}$,

$$\mathbb{E}[e^{\lambda X}] = 1 - q + q\cosh\left(\frac{\lambda}{\sqrt{q}}\right) = 1 + \sum_{k=1}^{\infty}\frac{\lambda^{2k}}{q^{k-1}(2k)!} \leq e^{\lambda^2/2} = 1 + \sum_{k=1}^{\infty}\frac{\lambda^{2k}}{2^k\,k!}$$

whenever $q \geq 1/3$. A sufficient condition for the inequality is that for all $k \geq 1$,

$$\frac{1}{q^{k-1}(2k)!} \leq \frac{1}{2^k\,k!} \iff q^{k-1} \geq \frac{2^k\,k!}{(2k)!} . \tag{8}$$

For $k = 1$ this is always true, for $k = 2$ it requires that $q \geq \frac{4 \times 2}{24} = \frac{1}{3}$. A simple induction shows that if $q \geq 1/3$, the condition is also satisfied for all $k \geq 3$. Reciprocally, if $q < 1/3$ then $\mathbb{E}[e^{\lambda X}] - e^{\lambda^2/2} \sim_{\lambda \to 0} -c\lambda^4$ for a positive constant $c$, and $P$ is not 1-sub-Gaussian. $\qquad\square$

This shows that Achlioptas' suggestion is the only "optimal" choice in terms of sparsity for a variance 1 and 1-sub-Gaussian distribution. Nevertheless, many other choices are possible, such as for example $P = \frac{1}{12}\delta_{-2} + \frac{1}{6}\delta_{-1} + \frac{1}{2}\delta_0 + \frac{1}{6}\delta_1 + \frac{1}{12}\delta_2$.

## 3 Very Sparse Random projections

We say that a random matrix with independent entries is $q$-sparse if each coefficient has probability at least $1 - q$ to be equal to zero. In the previous section, we showed that the minimal probability $q$ for the non-zeros values of a suitable 1-sub-Gaussian distribution $P$ is $1/3$. In fact, this result was proven in Li et al. (2006) with more complicated moment arguments. It allows to take a target dimension $d \geq 8\ln(n)/(\epsilon^2 - \epsilon^3)$ – see (2)– to get a $\epsilon$-quasi-isometry with non-zero probability, whatever the data $x$.

This does not exclude the possibility of using $q$-sparse projection matrices with $q < 1/3$, however. Technically speaking, the previous analysis remains quite conservative in that the sub-Gaussianity of $Z_i$ is deduced from the sub-Gaussianity of *each* of its summands. We may expect to gain a lot of sparsity by using the fact that a sum can be a lot more concentrated than each of its components. Figure 1 suggests that, at least under certain conditions on the data, much sparser matrices may be considered.

In this section, we present two results aimed at quantifying the minimum sparsity $q$ necessary to maintain the quasi-isometry condition with a dimension $d$ on the order of $\ln(n)/\epsilon^2$. In particular, Theorem 3 shows that $q$ can be as small as $\max_{i \neq j} \frac{\|x_i - x_j\|_\infty^2}{\|x_i - x_j\|_2^2}$. To finish, we establish that this is in fact a theoretical limit and that $q$ must be at least of this magnitude.

### 3.1 Towards maximal sparsity

Let $U \in \mathbb{R}^{d \times p}$ be a matrix of iid 1-sub-Gaussian entries with variance 1, and $\zeta \in \mathbb{R}^{d \times p}$ be a matrix of iid Bernoulli variables of parameter $q$ independent from $U$ that is used to *mask* a proportion $1 - q$ of the coefficients. We assume that for all $i, k$,

$$A_{ik} = \frac{1}{\sqrt{dq}}\zeta_{ik}U_{ik} . \tag{9}$$

For the coefficients of $U$, one can take Achlioptas' choice $\frac{1}{6}\delta_{-\sqrt{3}} + \frac{2}{3}\delta_0 + \frac{1}{6}\delta_{\sqrt{3}}$ to gain yet another fraction of sparsity on top of the mask. We apply the matrix $A$ to $n$ points $x_1, \ldots, x_n$ in a high-dimensional space

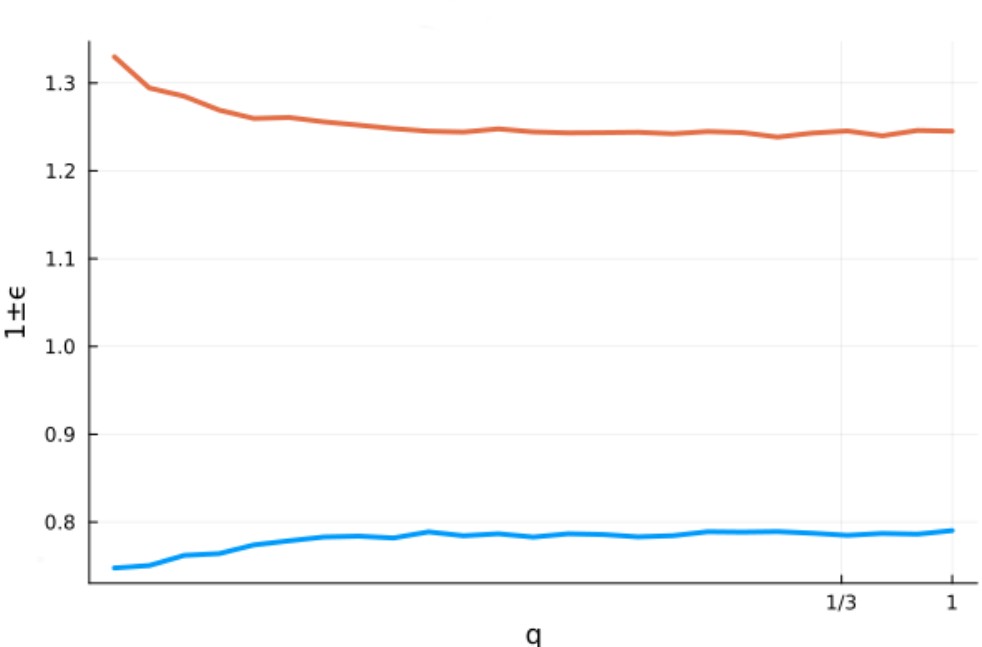

Figure 1: Admissible value $\epsilon$, in function of the sparsity parameter $q$ of the random projection entries. Each data point $x_i \in \mathbb{R}^{10000}$ has independent Gaussian entries. The projection matrix $A$ has independent entries with distribution $\frac{q}{2}\delta_{-\frac{1}{\sqrt{dq}}} + (1-q)\delta_0 + \frac{q}{2}\delta_{\frac{1}{\sqrt{qd}}}$. The target dimension is $d = 500$. The blue line shows $\min_{i,j} \frac{\|A(x_i - x_j)\|^2}{\|x_i - x_j\|^2}$, while the red line shows $\max_{i,j} \frac{\|A(x_i - x_j)\|^2}{\|x_i - x_j\|^2}$.
The value $q = 1/3$ seems to play no special role, much sparser matrices seem to respect pairwise distances just as well.

$\mathbb{R}^p$, and we look for the minimal conditions under which the quasi-isometry property (11) still holds with positive probability. We propose a first result in that direction.

**Theorem 2** *Let $x_1, \ldots, x_n \in \mathbb{R}^p$ and $\epsilon > 0$. For every $d \geq \dfrac{36 \ln\left(2n^2\right)}{\epsilon^2 - \epsilon^3}$ and every*

$$q \geq \max_{i \neq j} \frac{18 \|x_i - x_j\|_4^4 + 2\|x_i - x_j\|_\infty^2 \|x_i - x_j\|_2^2}{\epsilon^2 \|x_i - x_j\|_2^4} \ln(dn^2) , \tag{10}$$

*it holds with positive probability that*

$$\forall 1 \leq i, j \leq n, \quad (1 - \epsilon)\|Ax_i - Ax_j\|^2 \leq \|x_i - x_j\|^2 \leq (1 + \epsilon)\|Ax_i - Ax_j\|^2 . \tag{11}$$

In particular, Theorem 2 establishes that there exists a $q$-sparse matrix in $\mathcal{M}_{d,p}(\mathbb{R})$ satisfying the quasi-isometry condition if $d \gtrsim \ln(n)/\epsilon^2$ and $q \gtrsim \max_{i \neq j} \frac{\|x_i - x_j\|_\infty^2}{\epsilon^2 \|x_i - x_j\|_2^2} \ln(nd)$ up to a constant factor. Hence, if the coefficients of the differences $x_i - x_j$ for $i \neq j$ are of the same order of magnitude $1/\sqrt{p}$, then $q$ is allowed to be of order $\ln(nd)/(\epsilon^2 p)$, which is much smaller than $1/3$ if $p \gg 1/\epsilon^2$. The cost in terms of target dimension is only a multiplicative constant (that is not optimized in the previous reasoning). The proof of Theorem 2 relies on similar ideas as in the preceding section, and will be provided in Section A.

In Theorems 1 and 2, the target dimension is of order $\ln(n)/\epsilon^2$. The interest of Theorem 2 is mostly theoretical, as the constants can become pretty large. It turns out that if we allow slightly larger target dimensions of order $\mathrm{polylog}(n)/\epsilon^2$, then we can decrease even further the sparsity parameter $q$. For the sake of completeness, we state this version of the quasi-isometry property (11) in high-probability instead of just with positive probability.

**Theorem 3** *Let $x_1, \ldots, x_n$ be arbitrary vector in $\mathbb{R}^p$ and let $A \in \mathbb{R}^{d \times p}$ be a random matrix with independent entries $A_{ik} = \frac{1}{\sqrt{dq}} \zeta_{ik} U_{ik}$ with*

$$q \geq \max_{i \neq j} \frac{\|x_i - x_j\|_\infty^2}{\|x_i - x_j\|_2^2} . \tag{12}$$

*Then for any $\delta \in (0, 1)$ and any $d$ such that*

$$d \geq \frac{12}{\epsilon^2} \ln(3n/\delta) \left(1 + \sqrt{4\ln(nd/\delta)} + 2\ln(nd/\delta)\right)^2 , \tag{13}$$

*the $\epsilon$-quasi-isometry property (11) holds with probability at least $1 - \delta$.*

Up to a poly-logarithmic factor in $n$ and $\delta$, the minimal dimension $d_0$ satisfying Condition 13 is still of order $1/\epsilon^2$. The parameter $q$ can be chosen as small as $\max_{i \neq j} \frac{\|x_i - x_j\|_\infty^2}{\|x_i - x_j\|_2^2}$ while keeping the original guarantee of Johnson Lindenstrauss (11) with non-zero probability under the same condition (2) up to a poly-logarithmic factor: we require $d \geq d_0(n, 1, \epsilon)$ instead of $d \geq 8\ln(n)/(\epsilon^2 - \epsilon^3)$. In comparison to Theorem 2, we removed a factor of order $1/\epsilon^2$ in the minimal allowed sparsity $q$, at the cost of a poly-logarithmic factor in the target dimension.

## 3.2 About the sparsity conditions 12 and 10.

Condition (12), can be understood as a "not-too-high-sparsity" condition on the differences $x_i - x_j$, which we formalize as follows. For any constants $\kappa < \kappa' \in (0, 1)$ and integer any $s \in \{1, \ldots, p\}$, we say that a vector $v$ is $(\kappa, \kappa', s)$-full if $\|v\|_\infty \leq \sqrt{\kappa'/s}$ and if it has at least $s$ coordinates whose absolute value are at least equal to $\sqrt{\kappa/s}$, that is

$$\|v\|_\infty \leq \sqrt{\kappa'/s} \quad \text{and} \quad |\{k : |v_k| \geq \sqrt{\kappa/s}\}| \geq s . \tag{14}$$

This implies in particular that $\|v\|^2 \geq \kappa \geq \frac{\kappa}{\kappa'} s \|v\|_\infty^2$. Hence, if a set of vectors $\{x_1, \ldots, x_n\}$ is such that all the differences $x_i - x_j$ are $(\kappa, \kappa', s)$-full for $i \neq j$, then a sufficient condition implying (12) is

$$q \geq \left(\frac{\kappa'}{\kappa}\right) \frac{1}{s} . \tag{15}$$

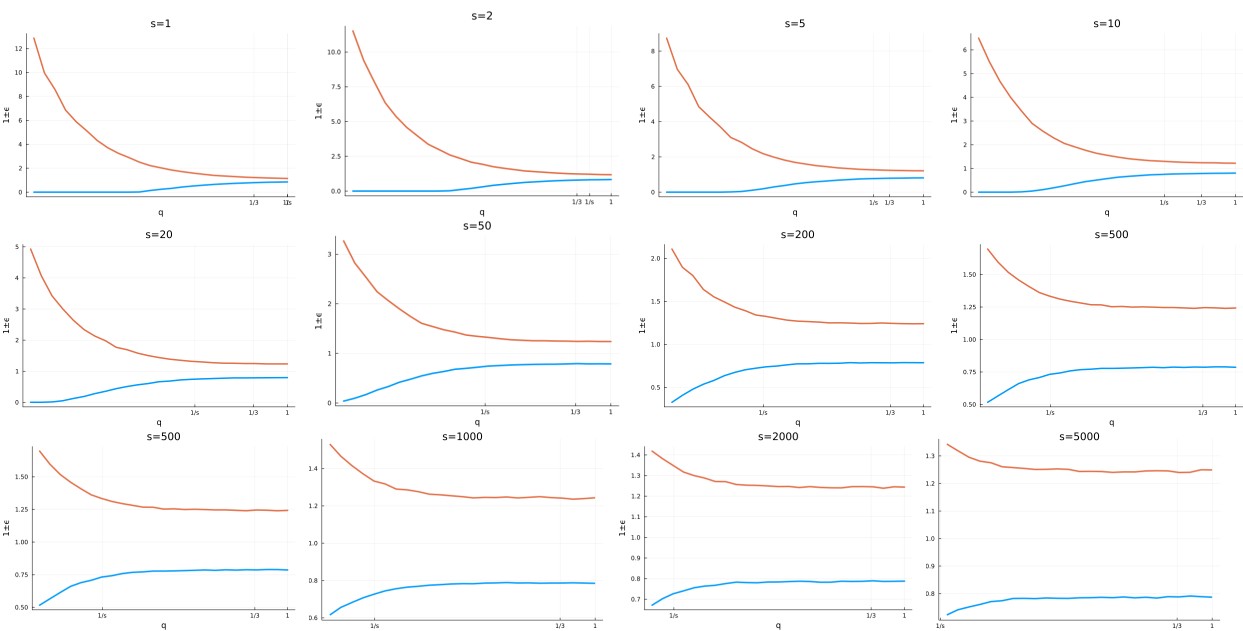

Figure 2: Admissible value $\epsilon$ in function of the sparsity parameter $q$ of the random projection entries (logarithmic scale), for different values of the sparsity $s$ of the data. Each data point $x_i \in \mathbb{R}^{10000}$ has exactly $s$ non-zero components, which are independent Gaussian entries. The coefficients of the projection matrix $A$ are independent and have distribution $\frac{q}{2}\delta_{-\frac{1}{\sqrt{dq}}} + (1-q)\delta_0 + \frac{q}{2}\delta_{\frac{1}{\sqrt{qd}}}$. The target dimension is $d = 500$. The blue line shows $\min_{i,j} \frac{\|A(x_i - x_j)\|^2}{\|x_i - x_j\|^2}$, while the red line shows $\max_{i,j} \frac{\|A(x_i - x_j)\|^2}{\|x_i - x_j\|^2}$. Observe that the scales of the ordinates are different between the plots.
It can be observed that quasi-isometry is ensured whenever $q \times s$ is sufficiently large.

In other words, we can take a matrix $A$ which has only a proportion $q \gtrsim 1/s$ of non-zero coefficients. This condition is for instance very weak in the dense case where the differences $x_i - x_j$ are $(\kappa, \kappa', p)$-full for all $i \neq j$, since it only requires $A$ to have a proportion non-zero coefficients of order $q \gtrsim 1/p$. In that case, all the coefficients of each difference $x_i - x_j$ are uniformly spread over the $p$ dimensions, in the sense that up to constants $\kappa, \kappa'$, $|x_{ik} - x_{jk}| \asymp 1/\sqrt{p}$ for any $k$.

Condition (12) becomes however much stronger when there exists a difference $x_i - x_j$ which is $s$-sparse for a small $s$, that is $|\{k : x_{ik} \neq x_{jk}\}| \leq s$. Indeed, in such a sparse case, $\|x_i - x_j\|_\infty^2 / \|x_i - x_j\|_2^2 \geq 1/s$, and the condition $q \geq 1/s$ is necessary to satisfy (12).

### 3.3 Theoretical limit to the sparsity

It turns out that the condition $q \gtrsim 1/s$ is in some sense optimal if we impose the dimension $d$ to be of order $1/\epsilon^2$ up to a poly-logarithm. This observation was already highlighted in Matoušek (2008), and we establish a result in this section that aligns with this perspective. Experimentally, Figures 2 and 3 dually confirm that random projections remain equally efficient as long as the proportion of non-zero coefficients is clearly above the minimum between $1/s$ and $1/3$. Our lower bound is based on the following intuition: let $y$ be $(1, 1, s)$-full vector, that is $\|y\| = 1$ and $y_i \in \{-1/\sqrt{s}, 1/\sqrt{s}, 0\}$, and let $S = \{k \in [p] : y_i \neq 0\}$. If $A \in \mathbb{R}^{d \times p}$ is any random matrix with independent coefficients such that $\mathbb{P}(A_{ik} \neq 0) \leq q$ for all $(i, k)$, then,

$$\mathbb{P}(Ay = 0) \geq \mathbb{P}\Big(\forall (i, k) \in [d] \times S, A_{ik} = 0\Big) = (1-q)^{ds} = e^{ds \ln(1-q)} \geq e^{-ds\frac{q}{1-q}} .$$

Hence, if $q \leq 1/(2ds)$, then $\mathbb{P}(Ay = 0) > 1/e$. In other words, there is no hope to satisfy the quasi-isometry property (11) with high probability if $q \leq 1/(2ds)$. This argument misses however the regime

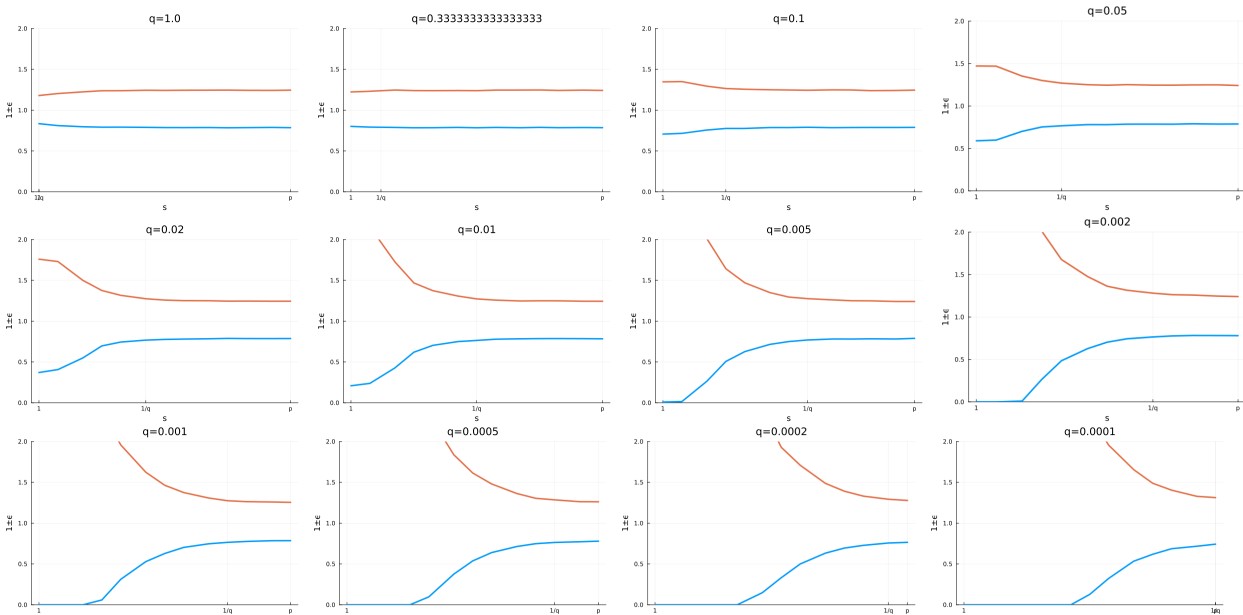

Figure 3: Admissible value $\epsilon$, in function on the sparsity parameter $s$ of the data (logarithmic scale), for different values of the sparsity $q$ of the projection matrix. Each data point of the $n = 100$ data point $x_i \in \mathbb{R}^{10000}$ has independent coefficients that are non-zero with probability $s/p$; the non-zero coefficients are independent and uniformly distributed on $\{-1, +1\}$. The projection matrix $A$ has independent entries with distribution $\frac{q}{2}\delta_{-\frac{1}{\sqrt{dq}}} + (1-q)\delta_0 + \frac{q}{2}\delta_{\frac{1}{\sqrt{qd}}}$. The target dimension is $d = 500$. The blue line shows $\min_{i,j} \frac{\|A(x_i - x_j)\|^2}{\|x_i - x_j\|^2}$, while the red line shows $\max_{i,j} \frac{\|A(x_i - x_j)\|^2}{\|x_i - x_j\|^2}$.

We observe that the values $q \geq 1/3$ ensure the quasi-isometry property whatever the data. For smaller values of $q$, the number $s$ of non-zero coefficients needs to be larger than $1/q$.

where $\epsilon^2/s \lesssim q \lesssim 1/s$ if $d \asymp 1/\epsilon^2$. The following theorem provides a general optimality result for all $q < 1/(240s)$, and hence fills the gap between $\epsilon^2/s$ and $1/s$ when $d \asymp 1/\epsilon^2$ up to a poly-logarithmic factor.

**Theorem 4** *Assume that $A \in \mathbb{R}^{d \times p}$ has iid coefficients distributed according to Achlioptas' distribution with parameter $q$ – see Eq. 7. Let $y \in \mathbb{R}^p$ be a unit vector with coordinates in $\{-1/\sqrt{s}, 1/\sqrt{s}, 0\}$. If $dqs\epsilon^2 \leq 1/2$, $qs < 1/240$, then*

$$\mathbb{P}(\|Ay\|^2 \in [1 - \epsilon, 1 + \epsilon]) \leq 1 - e^{-5000} \ .$$

In other words, if $d \asymp 1/\epsilon^2$ up to a polylog, then Theorem 4 only requires that $q \lesssim 1/s$ up to a polylog. We take a probability $1 - e^{-5000}$ that is very close to 1 in the theorem to match the two regimes where $dqs \gtrsim 1$ and $dqs \lesssim 1$. In the proof of Theorem 4, we also show that in the sub-case where $dqs \geq 1/2048$, the probability of success $\mathbb{P}(\|Ay\|^2 \in [1 - \epsilon, 1 + \epsilon])$ is smaller than $1/2$. The proof, which is given at the end of section A relies on the Tchebychev's inequality and on a control of the moments of order 2, 4, 6 and 8 of the random variable $\|Ay\|$.

## 4 En passant: concentration of non-negative quadratic forms

As a conclusion to this contribution, we highlight that the upper bound given in Section 3.1 is in fact strongly connected to the Hanson-Wright inequality for sub-Gaussian random variables – see e.g. Rudelson & Vershynin (2013), and Li (2024) for an application to the Johnson-Lindenstrauss lemma. This inequality is known with precise constants for quadratic form of Gaussian vectors – see Example 2.12 in Boucheron et al. (2013) – and it has been generalized with non-explicit constants to sub-Gaussian vectors, e.g. in Theorem 6.2.1 of Vershynin (2019). In the case where the quadratic form is assumed to be non-negative, the constants were established to be the same as in the Gaussian case in Hsu et al. (2012). For completeness, we propose a succinct proof of the Hanson-Wright inequality for sub-Gaussian vectors when the quadratic form is assumed to be non-negative.

A random vector $X \in \mathbb{R}^n$ is said to be 1-sub-Gaussian if, for any $u \in \mathbb{R}^d$, $\mathbb{E}[e^{u^T X}] \leq \exp\left(\|u\|_2^2/2\right)$. In particular, if $Z_1, \ldots, Z_d$ are independent real random variable and 1-sub-Gaussian, that is $\mathbb{E}[e^{\lambda Z_i}] \leq e^{\lambda^2/2}$, then for any orthogonal matrix $P$, $PZ$ is a 1-sub-Gaussian vector. In contrast to Rudelson & Vershynin (2013), we do not require in the following proposition the coordinates of $X$ to be independent.

**Proposition 3 (See also Theorem 2.1 of Hsu et al. (2012))** *Let $S$ be any $d \times d$ symmetric matrix with non-negative eigenvalues, and $X$ be a 1-sub-Gaussian vector. Then, for any $\ell \in [0, 1/(2\|S\|_{op}))$,*

$$\mathbb{E}_X[e^{\ell X^T SX}] \leq \exp\left(\ell \mathrm{Tr}(S) + \frac{\ell^2\|S\|_F^2}{1-2\ell\|S\|_{op}}\right) \ .$$

As a consequence of Proposition 3 and following the same computations as in Theorem 10 of Boucheron et al. (2013), it holds that with probability at least $1 - \delta$,

$$X^T SX \leq \mathrm{Tr}(S) + \sqrt{4\|S\|_F^2 \ln(1/\delta)} + 2\|S\|_{op} \ln(1/\delta) \ ,$$

for any $\delta \in (0, 1)$. In comparison to Theorem 6.2.1 of Vershynin (2019), the constants are explicit – they are indeed the same as in the Gaussian case. The argument applies however apply only to non-negative matrices. Proposition 3 can be deduced from the proof of Thereom 2.1 in Hsu et al. (2012) in the case $\mu = 0$ and $\sigma = 1$, but we still provide our own short proof as the underlying ideas are at the core of the upper bounds of this paper.

**Proof:** [Proof of Proposition 3] Let us write $S = P \mathrm{Diag}(\mu_1, \ldots, \mu_n) P^T$, where $\mu_1 \geq \cdots \geq \mu_n \geq 0$ and $P$ is an orthogonal matrix. Let also $Y$ be the sub-Gaussian vector equal to $PX$, and $\ell > 0$. By Fubini's theorem,

$$\mathbb{E}_X\left[e^{\ell X^T SX}\right] = \mathbb{E}_X\left[e^{\sum_{i=1}^n \ell\mu_i Y_i^2}\right] = \mathbb{E}_G\left[\mathbb{E}_X\left[e^{\sum_{i=1}^n \sqrt{2\ell\mu_i}Y_i G_i}\right]\right] \leq \mathbb{E}_G\left[e^{\sum_{i=1}^n \frac{1}{2}\ell\mu_i G_i^2}\right] \ ,$$

where $G_1, \ldots, G_n$ are independent standard and centered Gaussian random variables. Then,

$$E_G\left[e^{\sum_{i=1}^n \frac{1}{2}\ell\mu_i G_i^2}\right] = \prod_{i=1}^n \frac{1}{\sqrt{1-2\ell\mu_i}} = e^{\ell\mathrm{Tr}(S)}\prod_{i=1}^n \exp\left(-\tfrac{1}{2}\ln(1-2\ell\mu_i)-\ell\mu_i\right) \leq \exp\left(\ell\mathrm{Tr}(S)+\sum_{i=1}^n \frac{\ell^2\mu_i^2}{1-2s\mu_i}\right),$$

where the first inequality comes from the inequality $-\frac{1}{2}\ln(1-2\ell\mu_i)-\ell\mu_i = \int_0^\ell \frac{2s\mu_i}{1-2s\mu_i}ds \leq \frac{\ell^2\mu_i^2}{1-2\ell\mu_i}$. We conclude the proof by remarking that $\sum_{i=1}^n \frac{\ell^2\mu_i^2}{1-2s\mu_i} \leq \frac{\ell^2\|S\|_F^2}{1-2s\|S\|_{op}}$.

$\square$

### Acknowledgments

Aurélien Garivier thank Chaire SeqALO (ANR-20-CHIA-0020-01) and PEPR IA project FOUNDRY (ANR-23-PEIA-0003) for their support. The authors are thankful to Pierre Bellec, Nicolas Verzelen and the anonymous reviewers for their precious feedback.

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

# A  Proofs of Theorems 2, 3 and 4

While the proof of Theorem 2 remains as close as possible to that of Theorem 1, it provides some intuitions for the proof of Theorem 3.

## A.1  Proof of Theorem 2

Let $U \in \mathbb{R}^{d \times p}$ be a matrix of iid 1-sub-Gaussian entries with variance 1, and $\zeta \in \mathbb{R}^{d \times p}$ be a matrix of iid Bernoulli variables of parameter $q$ independent from $U$ that is used to *mask* a proportion $1 - q$ of the coefficients. We assume that for all $i, k$,

$$A_{ik} = \frac{1}{\sqrt{dq}} \zeta_{ik} U_{ik} \ , \tag{16}$$

and for any vector $y \in \mathbb{R}^p$ of norm $\|y\|_2 = 1$ write as before $Y = Ay$ and $Z_i = \sqrt{d} Y_i$.

Following the previous analysis, we need to bound $\mathbb{E}\big[\exp(\lambda Z_i^2)\big]$, and we know how to do it from $\mathbb{E}\big[\exp(\lambda Z_i)\big]$ when $Z_i$ is sub-Gaussian thanks to the argument of Inequality 3. Since $Z_i = \sum_{k=1}^{p} y_k \frac{\zeta_{ik} U_{ik}}{\sqrt{q}}$ is a sum of

many small contributions, for any fixed $\lambda$ we can bound $\ln \mathbb{E}\big[\exp(\lambda Z_i)\big]$ using only the local behavior of $\psi(\lambda) := \ln \mathbb{E}\big[\exp(\lambda U_{i,k})\big]$ around 0, which is of order $\lambda^2/2$ even when $\psi$ is not upper-bounded by that quantity. But using Inequality 3 would require a *uniform* control of $\psi$, which we cannot provide. We are hence obliged to take another path, by conditioning on the mask variables $(\zeta_{ik})$ and focusing on the "typical" behavior. Namely,

$$
\mathbb{E}\left[\exp\left(\lambda Z_i\right)\right] = \prod_{k=1}^{p} \mathbb{E}\left[\exp\left(\lambda y_k \frac{\sqrt{d}\zeta_{ik} U_{ik}}{\sqrt{dq}}\right)\right] = \prod_{k=1}^{p} \mathbb{E}\left[\mathbb{E}\left[\exp\left(\lambda \frac{y_k \zeta_{ik} U_{ik}}{\sqrt{q}}\right)\middle|\, \zeta_{i,1}, \ldots, \zeta_{i,k}\right]\right]
$$

$$
\leq \prod_{k=1}^{p} \mathbb{E}\left[\exp\left(\frac{\lambda^2 y_k^2 \zeta_{ik}}{2q}\right)\right] = \mathbb{E}\left[\exp\left(\frac{\lambda^2}{2}\sum_{k=1}^{p}\frac{y_k^2}{q}\zeta_{ik}\right)\right] .
$$

To upper bound this term, let for each $i \in \{1, \ldots, d\}$ and for $0 \leq \epsilon \leq 1$

$$
G_i = \left\{\left(1-\frac{\epsilon}{3}\right) \leq \sum_{k=1}^{p}\frac{y_k^2}{q}\zeta_{ik} \leq \left(1+\frac{\epsilon}{3}\right)\right\} .
$$

By Bernstein's inequality applied on the $\left[0, \frac{\|y\|_\infty^2}{q}\right]$-valued independent variables $\left(\frac{y_k^2}{q}\zeta_{ik}\right)_{1 \leq k \leq p}$, which have variance $\frac{y_k^4}{q^2} q(1-q) \leq \frac{y_k^4}{q}$,

$$
\mathbb{P}(\bar{G}_i) \leq 2\exp\left(-\frac{\epsilon^2/18}{\sum_{k=1}^{p}\frac{y_k^4}{q}+\frac{\|y\|_\infty^2 \epsilon}{9q}}\right) = 2\exp\left(-\frac{q\epsilon^2}{18\|y\|_4^4+2\|y\|_\infty^2}\right) ,
$$

which is smaller than $1/(dn^2)$ as soon as

$$
q \geq \frac{18\|y\|_4^4+2\|y\|_\infty^2}{\epsilon^2}\ln(dn^2) . \tag{17}
$$

On the event $G_i$, the behaviour of $Y_i$ is as expected:

$$
\mathbb{E}\left[\exp\left(\lambda Z_i\right)\mathbb{1}_{G_i}\right] \leq \exp\left(\frac{\lambda^2}{2}\left(1+\frac{\epsilon}{3}\right)\right)
$$

and $Z_i/\sqrt{1+\epsilon/3}$ is 1-sub-Gaussian. By Equation 3,

$$
\mathbb{E}\left[\exp\left(\frac{\ell Z_i^2}{1+\frac{\epsilon}{3}}\right)\mathbb{1}_{G_i}\right] \leq \frac{1}{\sqrt{1-2\ell}} .
$$

Left deviations may be treated similarly. Hence, on the event $G = \bigcap_{i=1}^{d} G_i$, the behaviour of $\|Y\|$ is just as in the non-sparse case: for all $\epsilon \leq 1$, by Equations 4 and 5,

$$
\mathbb{P}\big(G \cap \{\|Ay\|^2 \notin [1-\epsilon, 1+\epsilon]\}\big) \leq \mathbb{P}\left(G \cap \left\{\frac{1}{d}\sum_{i=1}^{d}\frac{Z_i^2}{1+\frac{\epsilon}{3}} > \frac{1+\epsilon}{1+\frac{\epsilon}{3}}\right\}\right) + \mathbb{P}\left(G \cap \left\{\frac{1}{d}\sum_{i=1}^{d}\frac{Z_i^2}{1-\frac{\epsilon}{3}} < \frac{1-\epsilon}{1-\frac{\epsilon}{3}}\right\}\right)
$$

$$
\leq \mathbb{P}\left(G \cap \left\{\frac{1}{d}\sum_{i=1}^{d}\frac{Z_i^2}{1+\frac{\epsilon}{3}} > 1+\frac{\epsilon}{3}\right\}\right) + \mathbb{P}\left(G \cap \left\{\frac{1}{d}\sum_{i=1}^{d}\frac{Z_i^2}{1-\frac{\epsilon}{3}} < 1-\frac{\epsilon}{3}\right\}\right)
$$

$$
\leq 2\,e^{-d\left(\frac{\epsilon^2-\epsilon^3}{36}\right)} .
$$

Consequently, denoting by $G^{i,j}$ the set $G$ corresponding to $y = \frac{x_i-x_j}{\|x_i-x_j\|}$,

$$
\mathbb{P}\left(\bigcup_{1 \leq i < j \leq n}\left\{\left\|A(x_i-x_j)\right\|^2 \notin \left[(1-\epsilon)\|x_i-x_j\|^2, (1+\epsilon)\|x_i-x_j\|^2\right]\right\}\right)
$$

$$
\leq \sum_{1 \leq i < j \leq n} P\big(\overline{G^{i,j}}\big) + \mathbb{P}\left(G \cap \left\{\frac{\|A(x_i-x_j)\|^2}{\|x_i-x_j\|^2} \notin [1-\epsilon, 1+\epsilon]\right\}\right) < \frac{1}{2} + n^2\,e^{-d\left(\frac{\epsilon^2-\epsilon^3}{36}\right)} \leq 1
$$

as soon as $q$ satisfies Eq.17 and $d \geq \frac{36 \ln\left(2n^2\right)}{\epsilon^2 - \epsilon^3}$.

## A.2  Proof of Theorem 3

Let $\odot$ be the Hadamard product, so that $A = \frac{1}{\sqrt{dq}} \zeta \odot U$. We assume that $y$ is unit vector of $\mathbb{R}^p$ representing one of the unit vector $\frac{x_i - x_j}{\|x_i - x_j\|}$, and we write as before $Y = Ay$. The coefficients of $w_i = \frac{1}{\sqrt{dq}} \zeta_i \odot y$ are equal to $w_{ik} = \frac{1}{\sqrt{dq}} \zeta_{ik} y_k$, and

$$Y_i^2 = \frac{1}{dq} \sum_{k',k} \zeta_{ik} \zeta_{ik'} U_{ik} U_{ik'} y_k y_{k'} = (U_{i\cdot}^T w_i)^2 \ .$$

**The upper bound**

$Y_i / \|w_i\|$ is 1-sub-Gaussian conditionally to $\zeta$. Hence, if $G$ is a standard Gaussian random variable, it holds conditionally to $\zeta$ that for any $\ell$ in $[0, 1/(2 \max \|w_i\|^2))$,

$$\mathbb{E}_U\left[e^{\ell Y_i^2}\right] = \mathbb{E}_G\left[\mathbb{E}_U\left[e^{\sqrt{2\ell} Y_i G}\right]\right] \leq \mathbb{E}_G\left[e^{\ell\|w_i\|^2 G^2}\right] = \frac{1}{\sqrt{1 - 2\ell\|w_i\|^2}} \leq \exp\left(\ell\|w_i\|^2 + \frac{\ell^2\|w_i\|^4}{1 - 2\ell\|w_i\|^2}\right) \ ,$$

where the last inequality comes from the fact that $-\frac{1}{2}\ln(1 - 2\ell\|w_i\|^2) - \ell\|w_i\|^2 = \int_0^\ell \frac{2s\|w_i\|^4}{1 - 2s\|w_i\|^2} ds \leq \frac{\ell^2\|w_i\|^4}{1 - 2\ell\|w_i\|^2}$. Hence, conditionally to $\zeta$, we have that

$$\mathbb{P}_\zeta(\|Ay\|^2 \geq 1 + \epsilon) \leq \mathbb{E}_\zeta\left[\exp\left(\ell \sum_{i=1}^d \|w_i\|^2 + \frac{d\ell^2 \max_i \|w_i\|^4}{1 - 2\ell \max_i \|w_i\|^2} - \ell(1 + \epsilon)\right)\right] \ . \tag{18}$$

Let us now integrate according to $\zeta$. The $w_{ik}$'s are independent, identically distributed random variables with law $\frac{y_k}{\sqrt{dq}} \mathcal{B}(q)$. Moreover, $\mathbb{Var}(w_{ik}^2) \leq \frac{y_k^4}{d^2 q^2} \mathbb{E}[\zeta_{ik}^4] \leq \frac{1}{d^2 q} y_k^4$. Bernstein's inequality together with a union bound over the $d$ possible indices $i = 1, \ldots, d$ gives that with probability at least $1 - \delta/(3n^2)$,

$$\max_i \|w_i\|^2 \leq \frac{1}{d} + \sqrt{2\frac{1}{d^2 q}\|y\|_4^4 \ln(3n^2 d/\delta)} + \frac{1}{dq}\|y\|_\infty^2 \ln(3n^2 d/\delta) \ .$$

The assumption (12) implies that $q \geq \|y\|_\infty^2$. Since $\|y\|^2 = 1$, $\|y\|_4^4 \leq \|y\|_\infty^2$ and the following event $G$ holds with probability at least $\delta/(3n^2)$:

$$G = \left\{\max_i \|w_i\|^2 \leq \frac{1}{d}\left(1 + \sqrt{4\ln(3nd/\delta)} + 2\ln(3nd/\delta)\right)\right\} = \left\{\forall i, \|w_i\|^2 \leq \frac{\Psi}{d}\right\} \ . \tag{19}$$

where for simplicity we write $\Psi = 1 + \sqrt{4\ln(3nd/\delta)} + 2\ln(3nd/\delta)$. Using the inequality $e^u \leq 1 + u + (e-2)u^2$ for any $u \in [0, 1]$, we have that for any $\ell \in [0, \frac{dq}{\|y\|_\infty^2})$,

$$\begin{aligned}
\mathbb{E}\left[\exp\left(\ell \sum_{i=1}^d \|w_i\|^2\right)\right] &= \prod_{i,k}\left(q\exp(\frac{\ell}{dq}y_k^2) + 1 - q\right) \\
&\leq \prod_{i,k}\exp\left(q(\exp(\frac{\ell}{dq}y_k^2) - 1)\right) \\
&\leq \exp(\ell + (e-2)\frac{\ell^2}{dq}\|y\|_4^4) \\
&\leq \exp(\ell + (e-2)\frac{\ell^2}{d}) \ .
\end{aligned}$$

Let us now integrate the conditional probability $\mathbb{P}_\zeta(\|Ay\|^2 \geq 1 + \epsilon)$ over $\zeta$. For any $\ell \in [0, d/(4\Psi))$, we have

$$\mathbb{P}(\|Ay\|^2 \geq 1 + \epsilon) \leq \mathbb{E}\left[\exp\left(\ell \sum_{i=1}^{d} \|w_i\|^2 + \frac{d\ell^2 \max_i \|w_i\|^4}{1 - 2\ell \max_i \|w_i\|^2} - \ell(1 + \epsilon)\right)\mathbf{1}_G\right] + \frac{\delta}{3n^2}$$

$$\leq \mathbb{E}\left[\exp\left(\ell \sum_{i=1}^{d} \|w_i\|^2 + 2\ell^2 \frac{A^2}{d} - \ell(1 + \epsilon)\right)\right] + \frac{\delta}{3n^2}$$

$$\leq \exp\left((e - 2)\frac{\ell^2}{d} + 2\ell^2 \frac{\Psi^2}{d} - \ell\epsilon\right) + \frac{\delta}{3n^2}$$

The second inequality comes from Equation 18, which holds true under the event $G$ defined in (19). The third inequality comes from the fact that $\ell \leq d/(4\Psi^2) \leq d \leq dq/\|y\|_\infty^2$ and the above upper bound on $\mathbb{E}[\exp(\ell \sum_{i=1}^{d} \|w_i\|^2)]$.

Choosing $\ell = d\epsilon/(2(e - 2) + 4\Psi^2) \leq d/(4\Psi^2)$, we get

$$\mathbb{P}(\|Ay\|^2 \geq 1 + \epsilon) \leq \exp\left(-\frac{d\epsilon^2}{2(e - 2) + 4\Psi^2}\right) + \frac{\delta}{3n^2}.$$

Hence, if $d \geq d_0 = 12\ln(3n/\delta)\Psi^2/\epsilon^2$, we obtain that

$$\mathbb{P}(\|Ay\|^2 \geq 1 + \epsilon) \leq \exp\left(-2\ln\frac{3n}{\delta}\right) + \frac{\delta}{3n^2} \leq \frac{2\delta}{3n^2}.$$

A union bound all the $n(n - 1)/2 \leq n^2$ pairs gives that

$$\mathbb{P}\left(\bigcup_{1 \leq i < j \leq n} \left\{\|A(x_i - x_j)\|^2 \geq (1 + \epsilon)\|x_i - x_j\|^2\right\}\right) \leq 2\delta/3. \tag{20}$$

**The lower bound**

For the lower bound, we use the same arguments as in section 2.1.2. We still have that $\mathbb{E}[(Ay)_i^2] = 1/d$ for any $i \in \{1, \ldots, d\}$, but we since the variables $(Ay)_i$ are not sub-Gaussians, do not have the bound $\mathbb{E}[(Ay)_i^4] \leq 3$. Instead, we bound the fourth moment as follows:

$$\mathbb{E}[(Ay)_i^4] \leq \frac{3}{d^2q^2} \sum_{k \neq k'} q^2 y_k^2 y_{k'}^2 + \frac{1}{d^2q^2} \sum_{k=1}^{d} q y_k^4 \mathbb{E}[U_{ik}^4]$$

$$\leq \frac{3}{d^2} + \frac{3\|y\|_\infty^2}{d^2q} \leq \frac{6}{d^2}.$$

Hence,

$$\mathbb{P}\left(\|Ay\|^2 \leq 1 - \epsilon\right) \leq \exp\left(d\ln\left(1 - \frac{\ell}{d} + 3\frac{\ell^2}{d^2}\right) + \ell(1 - \epsilon)\right) \leq \exp\left(3\frac{\ell^2}{d} - \ell\epsilon\right).$$

Choosing $\ell = \epsilon/6$, we obtain that

$$\mathbb{P}(\|Ay\|^2 \leq 1 - \epsilon) \leq \exp(-\frac{d\epsilon^2}{12}).$$

If $d \geq d_0 \geq 24\ln(3n/\delta)$, then we obtain

$$\mathbb{P}(\|Ay\|^2 \leq 1 - \epsilon) \leq \delta/(3n^2).$$

Hence, from a union bound over the at most $n^2$ possible pairs $x_i, x_j$, we obtain that

$$\mathbb{P}\left(\bigcup_{1 \leq i < j \leq n} \left\{\|A(x_i - x_j)\|^2 \leq (1 - \epsilon)\|x_i - x_j\|^2\right\}\right) \leq \delta/3 . \tag{21}$$

We conclude from the upper bound (20) and the lower bound (21) that if $q \geq \max_{i \neq j} \frac{\|x_i - x_j\|_\infty^2}{\|x_i - x_j\|_2^2}$ and if $d \geq d_0(n, \delta, \epsilon)$, the $\epsilon$-quasi isometry property (11) holds with probability at least $1 - \delta$, that is

$$\mathbb{P}\left(\bigcup_{1 \leq i < j \leq n} \left\{\|A(x_i - x_j)\|^2 \notin \left[(1 - \epsilon)\|x_i - x_j\|^2, (1 + \epsilon)\|x_i - x_j\|^2\right]\right\}\right) \leq 2\delta/3 + \delta/3 \leq \delta .$$

### A.3 Proof of Theorem 4

Let $y$ be a unit vector of $\mathbb{R}^p$. If $dqs \leq 1/2048$, then we have that

$$\mathbb{P}(\|Ay\|^2 \notin [1 - \epsilon, 1 + \epsilon]) \geq \mathbb{P}(Ay = 0) \geq e^{-ds\frac{q}{1-q}} \geq e^{-5000} ,$$

which proves the result in that case.

In what follows, we assume that $dqs \geq 1/2048$. Chebychev's inequality implies that

$$\mathbb{P}(\|Ay\|^2 \in [1 - \epsilon, 1 + \epsilon]) = \mathbb{P}((\|Ay\|^2 - 1)^2 \leq \epsilon^2)$$
$$\leq \frac{\mathbb{V}\text{ar}\left[(\|Ay\|^2 - 1)^2\right]}{\mathbb{E}\left[(\|Ay\|^2 - 1)^2\right] - \epsilon^2} .$$

Subsequently, we give a lower bound of $\mathbb{E}[(\|Ay\|^2 - 1)^2]$ and an upper bound of $\mathbb{V}\text{ar}\left[(\|Ay\|^2 - 1)^2\right]$. We denote by $X$ a random variable following the distribution of one coefficients of $A$. $X$ can be written $\frac{1}{\sqrt{dq}}\zeta U$, where $\zeta \sim \text{Bern}(q)$ an $U \sim \mathcal{U}(\{-1, 1\})$ are independent. It holds in particular that for any $k \geq 1$, $\mathbb{E}[X^{2k+1}] = 0$ and $\mathbb{E}[X^{2k}] = \frac{1}{d^k q^{k-1}}$.

**Lower bound of $\mathbb{E}[(\|Ay\|^2 - 1)^2]$.**

$$\mathbb{E}\left[(\|Ay\|^2 - 1)^2\right] = \mathbb{E}\left[\|Ay\|^4\right] - 1 = \mathbb{E}\left[\left(\sum_{i=1}^{d}\sum_{k=1}^{p}\sum_{l=1}^{p} A_{ik}A_{il}y_k y_l\right)^2\right] - 1$$

$$= \sum_{i_1, i_2, k_1, k_2, l_1, l_2} \mathbb{E}\left[\prod_{u \in \{1,2\}} A_{i_u k_u} A_{i_u l_u} y_{k_u} y_{l_u}\right] - 1 ,$$

where the final sum is over all $(i_1, i_2) \in [d]^2$ and all $(k_1, k_2, l_1, l_2) \in [p]^4$. Let us fix $i_1, i_2$ such that $i_1 = i_2$. Since $\mathbb{E}[A_{ik}] = \mathbb{E}[A_{ik}^3] = 0$ for any $i, k$, either $k_1 = k_2 = l_1 = l_2$ or there is exactly two pairs of equal indices among $(k_1, k_2, l_1, l_2)$. Since there are exactly 3 possible ways of matching 2 pairs among the four indices, we have that

$$\sum_{k_1, k_2, l_1, l_2} \mathbb{E}\left[\prod_{u \in \{1,2\}} A_{i_u k_u} A_{i_u l_u} y_{k_u} y_{l_u}\right] - 1 = \mathbb{E}[X^4]\|y\|_4^4 + 3\mathbb{E}[X^2]^2 \left(\|y\|_2^4 - \|y\|_4^4\right) = \frac{1}{d^2 qs} + \frac{3}{d^2}\left(1 - \frac{1}{s}\right) .$$

If $i_1 \neq i_2$, then we necessarily have that $k_1 = l_1$ and $k_2 = l_2$ for non-zero contributions. Hence, in that case,

$$\sum_{k_1, k_2, l_1, l_2} \mathbb{E}\left[\prod_{u \in \{1,2\}} A_{i_u k_u} A_{i_u l_u} y_{k_u} y_{l_u}\right] - 1 = \mathbb{E}[X^2]^2\|y\|_2^4 = \frac{1}{d^2} .$$

Combining the two cases, we obtain that

$$\mathbb{E}\left[(\|Ay\|^2 - 1)^2\right] = \frac{d}{d^2qs} + \frac{3d}{d^2}\left(1 - \frac{1}{s}\right) + \frac{d(d-1)}{d^2} - 1 \geq \frac{1}{dqs} \ . \tag{22}$$

**Upper bound of** $\mathbb{V}\mathrm{ar}\left[\|Ay\|^2 - 1)^2\right]$.

$$\mathbb{V}\mathrm{ar}\left[(\|Ay\|^2 - 1)^2\right] \leq \mathbb{E}\left[(\|Ay\|^2 - 1)^4\right] = \mathbb{E}\left[\|Ay\|^8\right] - 4\mathbb{E}\left[\|Ay\|^6\right] + 6\mathbb{E}\left[\|Ay\|^4\right] - 4\mathbb{E}\left[\|Ay\|^2\right] + 1$$
$$\leq \mathbb{E}\left[\|Ay\|^8\right] - 4\mathbb{E}\left[\|Ay\|^6\right] + \frac{6}{dqs} + 3 + \frac{18}{d} \ .$$

The inequality comes from the above computation of $E[\|Ay\|^4]$. In what follows, we first upper-bound $\mathbb{E}[\|Ay\|^8]$ and then we lower-bound $\mathbb{E}\left[\|Ay\|^6\right]$. For the latter, the idea is to cancel out the terms of constant order or of order $1/(dqs)$. Following the same lines as in the computation of $\mathbb{E}[\|Ay\|^4]$, we observe that

$$\mathbb{E}\left[\|Ay\|^8\right] = \sum_{(i_u),(k_u),(l_u)} \mathbb{E}\left[\prod_{u\in\{1,2,3,4\}} A_{i_u k_u} A_{i_u l_u} y_{k_u} y_{l_u}\right] \ , \tag{23}$$

where the sum is over all $((i_u)_{u=1,\dots,4}, (k_u)_{u=1,\dots,4}, (l_u)_{u=1,\dots,4}) \in [d]^4 \times [p]^8$. Let us consider the following sets for the indices $(i_u)$:

1. $(i_u) \in I_1$ if the $i_u$'s are pairwise distinct. in that case, $|I_1| = d(d-1)(d-2)(d-3) \leq d^4$

2. $(i_u) \in I_2$ if there are exactly two equal indices among the $i_u$'s. In other words, $(i_u)$ is a permutation of $(i, i, i', i'')$ where $i$, $i'$, $i''$ are pairwise distinct. Here, $|I_2| = 6d(d-1)(d-2) \leq 6d^3$

3. $(i_u) \in I_3$ if there are exactly three equal indices among the $i_u$'s, i.e $(i_u)$ is a permutation of $(i, i, i, i')$ where $i \neq i'$. Here, $|I_3| = 4d(d-1) \leq 4d^2$

4. $(i_u) \in I_4$ if all the $i_u$'s are equal. Here, $|I_4| = d$

5. $(i_u) \in I_5$ if there are exactly two pairs of equal indices among the $i_u$'s. Here, $|I_5| = 3d(d-1) \leq 3d^2$

The sets $(I_v)$ are disjoint, and the reader can check that the sum of their sizes is equal to $d^4$. Let us fix $(i_u) \in [d]^4$, and consider the five following cases, each corresponding to one of the sets $(I_v)$.

1. If $(i_u) \in I_1$, then the expectation of the product over $u \in \{1,2,3,4\}$ is non-zero only if $k_u = l_u$ for all $u \in \{1,2,3,4\}$. Hence,

$$\sum_{(k_u),(l_u)} \mathbb{E}\left[\prod_{u\in\{1,2,3,4\}} A_{i_u k_u} A_{i_u l_u} y_{k_u} y_{l_u}\right] = \frac{1}{d^4}\|y\|_2^8 = \frac{1}{d^4} \ .$$

2. If $(i_u) \in I_2$, we assume that without loss of generality that $(i_1, i_2, i_3)$ are pairwise distinct and that $i_3 = i_4$. In that case, we have a non-zero contribution only if $k_1 = l_1$, $k_2 = l_2$ and if either $(k_3 = l_3 = k_4 = l_4)$ or there are two matching pairs among the indices $(k_3, l_3, k_4, l_4)$ (3 possible matching). Hence, using the fact that $\|y\|_2 = 1$ and $\|y\|_4^4 = 1/s$:

$$\sum_{(k_u),(l_u)} \mathbb{E}\left[\prod_{u\in\{1,2,3,4\}} A_{i_u k_u} A_{i_u l_u} y_{k_u} y_{l_u}\right] \leq \frac{1}{d^4 q}\|y\|_2^4\|y\|_4^4 + \frac{3}{d^4}\|y\|_2^8 = \frac{1}{d^4 qs} + \frac{3}{d^4} \ .$$

3. If $(i_u) \in I_3$, we assume that $i_1, i_2$ are distinct and that $i_2 = i_3 = i_4$. In that case, we have a non-zero contribution if $k_1 = l_1$ and if either $(k_2 = l_2 = k_3 = l_3 = k_4 = l_4)$ or if there are 3 matching pairs among $(k_1, l_2, k_3, l_3, k_4, l_4)$ $(5 \cdot 3 = 15$ possible matchings). Hence, using also that $qs \leq 1$,

$$\sum_{(k_u),(l_u)} \mathbb{E}\left[\prod_{u \in \{1,2,3,4\}} A_{i_u k_u} A_{i_u l_u} y_{k_u} y_{l_u}\right] \leq \frac{1}{d^4 q^2} \|y\|_2^2 \|y\|_6^6 + \frac{15}{d^4} \|y\|_2^8 \leq \frac{16}{d^4 q^2 s^2} \ .$$

4. If $(i_u) \in I_4$, then there is a non-zero contribution in one of the three following cases. Either the $k_u$'s and $l_u$'s are all equal, or there are 2 groups among the $k_u$'s and $l_u$'s, each made of 4 indices that are all equal $(\frac{1}{2}\binom{8}{4} = 35$ possibilities), or there are 4 matching pairs $(7 \cdot 5 \cdot 3 = 105$ possible matching). Hence,

$$\sum_{(k_u),(l_u)} \mathbb{E}\left[\prod_{u \in \{1,2,3,4\}} A_{i_u k_u} A_{i_u l_u} y_{k_u} y_{l_u}\right] \leq \frac{1}{d^4 q^3} \|y\|_8^8 + \frac{35}{d^4 q^2} \|y\|_4^8 + \frac{105}{d^4} \|y\|_2^8 \leq \frac{141}{d^4 q^3 s^3} \ .$$

5. If $(i_u) \in I_5$, assume without loss of generality that $i_1 = i_2$, $i_3 = i_4$ and $i_2 \neq i_3$. Then there are two possibilities for each pairs $(i_1, i_2)$ and $(i_3, i_4)$. Either $k_1 = l_1 = k_2 = l_2$ (resp. $k_3 = l_3 = k_4 = l_4$) or there are three pairs of equal indices among $k_1, l_1, k_2, l_2$ (resp. $k_3, l_3, k_4, l_4$). This gives

$$\sum_{(k_u),(l_u)} \mathbb{E}\left[\prod_{u \in \{1,2,3,4\}} A_{i_u k_u} A_{i_u l_u} y_{k_u} y_{l_u}\right] \leq \left(\frac{1}{d^2 q} \|y\|_4^4 + \frac{3}{d^2} \|y\|_2^4\right)^2 \leq \frac{16}{d^4 q^2 s^2} \ .$$

Decomposing the equation (23) into these five above cases and using the assumption $dqs \geq 1$, we obtain that

$$\mathbb{E}\left[\|Ay\|^8\right] \leq 1 + \frac{6}{dqs} + \frac{18}{d} + \frac{4*16}{d^2 q^2 s^2} + \frac{141}{d^3 q^3 s^3} + \frac{3*16}{d^2 q^2 s^2} \leq 1 + \frac{6}{dqs} + \frac{18}{d} + \frac{253}{d^2 q^2 s^2} \ ,$$

which implies that

$$\mathbb{V}\text{ar}\left[(\|Ay\|^2 - 1)^2\right] \leq \mathbb{E}[\|Ay\|^8] - 4\mathbb{E}[\|Ay\|^6] + \frac{6}{dqs} + 3 + \frac{12}{d}$$

$$\leq 4 + \frac{12}{dqs} - 4\mathbb{E}[\|Ay\|^6] + \frac{253}{d^2 q^2 s^2} + \frac{30}{d} \ .$$

We now show that the term $4 + \frac{12}{dqs}$ is smaller than $4\mathbb{E}[\|Ay\|^6]$. Doing the same reasoning as above, we can write

$$\mathbb{E}\left[\|Ay\|^6\right] = \sum_{(j_u),(k_u),(l_u)} \mathbb{E}\left[\prod_{u \in \{1,2,3\}} A_{i_u k_u} A_{i_u l_u} y_{k_u} y_{l_u}\right] \ , \tag{24}$$

where the sum is over all $(j_u), (k_u), (l_u)$ in $[d]^3 \times [p]^6$. The product is always non-negative, and we consider the sets

$$J_1 = \{(j,j,j) : \ j \in [d]\} \text{ and } J_2 = \{(j_1, j_2, j_3) : \ \text{two of the } j_u \text{ are equal and distinct from the other one}\} \ .$$

We have that $|J_1| = d^3$ and $|J_2| = 3d(d-1)$, so that

$$\mathbb{E}\left[\|Ay\|^6\right] \geq \sum_{(j_u) \in J_1,(k_u),(l_u)} \mathbb{E}\left[\prod_{u \in \{1,2,3\}} A_{i_u k_u} A_{i_u l_u} y_{k_u} y_{l_u}\right] + \sum_{(j_u) \in J_2,(k_u),(l_u)} \mathbb{E}\left[\prod_{u \in \{1,2,3\}} A_{i_u k_u} A_{i_u l_u} y_{k_u} y_{l_u}\right]$$

$$= \|y\|_2^6 + 3d(d-1)\frac{1}{d^3 q} \|y\|_4^4 \|y\|_2^2$$

$$= 1 + \frac{3}{dqs} - \frac{3}{d^2 qs} \geq 1 + \frac{3}{dqs} - \frac{3}{d^2 q^2 s^2}$$

To conclude, we obtain

$$\mathbb{Var}\left[(\|Ay\|^2 - 1)^2\right] \leq 4 + \frac{12}{dqs} - 4\mathbb{E}[\|Ay\|^6] + \frac{253}{d^2q^2s^2} + \frac{30}{d}$$
$$\leq \frac{256}{d^2q^2s^2} + \frac{30}{d} \ .$$

Combining this latter upper bound with (22), we conclude that

$$\mathbb{P}\big(\|Ay\|^2 \in [1 - \epsilon, 1 + \epsilon]\big) \leq \frac{\frac{256}{d^2q^2s^2} + \frac{30}{d}}{\frac{1}{dqs} - \epsilon^2} \leq \frac{\frac{256}{dqs} + 30qs}{1 - dqs\epsilon^2} \leq 1/2 \ ,$$

where we used in the last inequality the assumption that $dqs\epsilon^2 \leq 1/2$, $qs \leq 1/240$ and $dqs \geq 2048$. This concludes the proof of Theorem 4.

