# OpenReview forum: "On Sparsity and Sub-Gaussianity in the Johnson- Lindenstrauss Lemma"
_TMLR — Accepted by TMLR_

### Review · Reviewer_J33z · 2025-07-01

**Summary Of Contributions:**

The authors improve, simplify, and consolidate some of the proofs of the Johnson-Lindenstrauss lemma, and show that the statements which were made for the Gaussian case extend to sub-Gaussian entries effectively without modification, with matching constants. Along the way they propose several new proof techniques which simplify derivations, while always referring to primary references for ideas and bits and pieces. In the first part they explain the mechanics behind standard proofs and introduce their techniques; in the second part they zoom in on sparse random projections and the interplay between maximum sparsity of data / matrices, generalizing and improving the results of Achlioptas and others.

**Audience:**

Yes

**Broader Impact Concerns:**

no concerns

**Claims And Evidence:**

Yes

**Requested Changes:**

- please address my suggestions about improving comparisons with background / making it explicit how and why the new arguments simplify things
- please correct the numerous typos

**Strengths And Weaknesses:**

I am not an expert on the latest and greatest on Johnson-Lindenstrauss or sub-Gaussian concentration, but I've been a user of both and I understand standard proof arguments.

With this background stated, I think this is a very nice paper:

- it clearly states its contribution
- it clearly delineates what is known, without overselling or underselling, pleasure to read in 2025
- the arguments are clear
- I checked the math somewhat carefully and derivations and proofs seem correct

I'm putting here a mixed list of comments and suggestions. There is nothing weak about the paper. The main suggestion for improvement is to do more signposting and explain (for non-experts) _how_ the proof mechanics are different from earlier approaches (and thus put 2-3 sentence about these earlier proofs), or at least what is the key new idea / trick / insight.

- When you write things like "the central argument is simplified with a new approach" it would be nice to state the gist of the new approach and how it's different from the old approach. This could be done here or elsewhere, but the same comment applies to most of the manuscript. Rather than learning that some earlier argument (e.g. Achlioptas) is "pretty involved", I'd prefer to get some quick understanding of the mechanics of difference between the two arguments.

- (3) is exactly the same as the previous display. I've no problem with that but I was wondering if it's just for didactic purposes or there's something we should note. If I were to keep one, it'd be the second one for its elegance.

- on page 3, "the concave function l \mapsto l(1+eps) - ln E[e^{lX}] = l(1+eps)...": should X be squared in the exponent? (I understand that it is meant as a dummy rv but it may be simpler to follow if you put Z here.) Should = be an inequality (from (3))?

- btw, here you use both ln and log to refer to the same function
- proof of Proposition 2: mention that \zeta is independent of U

- I understand the theoretical value of Theorem 2, but I'll point out that the dimension can get quite huge. E.g. for n = 10000, eps = 0.1, one gets that d should be at least ~75000. With eps = 0.1 to get d = n/100 one needs n about 1.3e7.

- About Figure 1: I'm wondering about quantifiers. All statements of J-L lemma work for any set of points. If we look at a "universal" random A that works with prob at least 1-delta (rather than saying, for a set of points, there is a matrix), then one could wonder about "adversarial" point sets when q gets very small, which would be impossible to discover by generating random points.

- Second display on p11 (in Theorem 4): is the inequality pointing in the wrong direction?

### Typos and typesetting

- page 1: "Property equation 11 is satisfied" - better to refer to eq (1) on the same page
- the macro you use to reference equations results in repetition (Equation equation (x)); same for "Property equation 11" which sounds weird (there's also Eq. equation 7, etc)
- a similar comment for citation ("... e.g. was established by Matousek Matousek (2008)")
- "on streams where then devised"; where -> were
- "which is done is the two following" -> in the two following
- "... bounded by those of the standard Gaussian. and to ...": remove full stop
- proof of Theorem 1: "where the (T_{i,j}) centered" -> are centered
- page 2: mutliplicative constant -> multiplicative constant
- several places, Archlioptas -> Achlioptas
- why is there an inline QED sign in the first paragraph of page 6? It would be nicer to open a new paragraph.
- first display on p11: "(i, k) \in d \times S" - it's clear what you mean but please use precise notation (e.g. [d] \times [S], with proper definitions)
- p11: Gaussian chaos -> Gaussian case?

---

> ### Author Response · Authors · 2025-08-05
> **response**
>
> We thank all three reviewers for their positive feedback and constructive comments. We tried our best to improve the paper thanks to their numerous suggestions.
>
> * Regarding the necessity to  state the gist of the new approach and how it's different from the old approach: We have added in the early parts of this revisions a few precisions on the mentioned approaches (e.g. moment arguments for Archiloptas) and on our approach. We have also reformulated a few references (e.g. on p.1) to avoid confusion. Additionally, we have included a comparison with Li's paper at the location you mentioned.
>
> * We indeed could not resist the temptation to provide two proofs of Equation (3), for pedagogical reasons. While the first one is pretty close to [Wainwright] and was inspired to us by this book, we find the second approach (which we did not find anywhere) somewhat more elegant, and the connection between them appears to us as interesting curiosity.
>
> * Thank you for pointing out the typos in our maximization of the concave function.
>
> * Sorry for that: we have replaced all instances of "log" with "ln" throughout the document for consistency.
>
> * In the proof of Proposition 2: as pointed out, we added the precision that $\zeta$ is independent of $U$.
>
> * In Theorem 2, following your remark we also noted that the constants can get quite huge, and that the interest of the theorem is mostly theoretical.
>
> * Regarding your question about adversarial points, the Johnson-Lindenstrauss theorem succeeds precisely because such adversarial points are absent in typical scenarios, even for non-sparse cases. The "worst-case" adversarial point $x$ satisfies $\|Ax\|_2^2 = \|AA^T\|_{op}$, which has order $d$ when $d=p$ (and grows larger when $d \leq p$). Since this quantity exceeds $1$ by far, the matrix $A$ cannot maintain quasi-isometric properties for this particular $x$. This is closely related to the diameter of the semicircle in the semicircle law.
> An alternative perspective on adversarial points emerges through sphere covering arguments. If we require the JL inequality to hold for every possible adversarial point, we must choose $n$ proportional to the covering size, which scales as $2^p$. This yields $\log(n) \asymp p$, eliminating any dimension reduction benefit. For the JL lemma to fail, an adversary should select a point with high correlation to the leading eigenvector $x$ of $A$.
>
> * If we are not mistaken, the inequality on page 11 points in the right direction, with high probability (at least $1-\delta$). It says that $X^TSX$ is below some upper confidence bound.
>
> * Thank you for the comprehensive list of typographical errors, which we have addressed accordingly. Regarding Gaussian Chaos, this term refers to a quadratic form of a Gaussian vector, with the Hanson-Wright inequality being the standard method for establishing control over such expressions. For greater clarity, we have replaced "Gaussian Chaos" with the more descriptive term "quadratic form of a Gaussian vector".

---

> > ### Comment · Reviewer_J33z · 2025-09-03
> >
> > Thank you for your earnest engagement with my comments. In the meantime I got educated on Gaussian chaos!

---

### Review · Reviewer_nbQE · 2025-07-03

**Summary Of Contributions:**

This paper presents mathematical theorems improving the classical JL theorem. JL is a fundamental result that justifies dimensionality reduction methods using linear projections (a.k.a linear sketching). In particular, this paper predicts existence of sparser projection matrices than the ones the original theorem predicts, provided that the data is not too sparse. The paper also provides a lower bound on sparsity in the sense that too sparse random projection matrices, being at the core of the JL argument, have a non-vanishing probability of failure.

**Audience:**

Yes

**Broader Impact Concerns:**

No ethical implications, the contribution is completely mathematical.

**Claims And Evidence:**

Yes

**Requested Changes:**

The contributions of the paper in the introduction remain ambiguous. It is good to list the contributions in the introduction.

I am not sure why the authors decide to present proofs of standard results (propositions) in the body,  while Theorem 2-4 are the actual contributions. It makes more sense to me to have a sketch of the proofs for these results in the body.

The relation of Section 4 to the rest of the paper is not clear to me. Please clarify.

**Strengths And Weaknesses:**

The result of the paper is a progress in the theory of random projections. Sparse matrices are appealing for dimensionality reduction as they impose remarkably lower computational effort. The paper is well-written and easy to understand.

Despite mathematical contributions, the connection of the paper to ML is not well justified. More examples from ML can be helpful to make this connection.

A limitation of theorem 4, if I understood it correctly, is that it does not rule out existence of a suitable projection for sparse matrices, which would be a proper converse theorem to JL. At best, the result shows that finding such a matrix is not an easy task anymore.

---

> ### Author Response · Authors · 2025-08-05
> **response**
>
> We thank all three reviewers for their positive feedback and constructive comments. We tried our best to improve the paper thanks to their numerous suggestions.
>
> * We agree that this contribution is mostly theoretical: we believe that it can help the community to understand the theoretical analysis of random projections, especially with non-gaussian entries and in the sparse case.
>
> * We also agree that our Theorem 4 is more a converse to the probabilistic argument that yields the Johnson-Lindenstrauss lemma, and not really to the JL existence result. We present it in the paper as an optimality result on the sparsity required by a random matrix to be a good projection matrix with high probability.
>
> * We tried to reformulate the introduction to make the two points above more clear. We also reformulated the contribution part and the presentation of the content so as to clarify the list of contributions.
>
> * Regarding the breakdown of the proofs between the main text and the appendix: we tried to include in the main text all the key ideas, but left the most technical parts for the motivated readers in the appendix. We tried to sketch the ideas behind the proofs of Theorems 2-4. We do believe that the proofs included in the main text, even if they lead to already known results, are of interest to the community by their originality and clarity, and a useful preparation before reading the appendix.
>
> * We have modified and hopefully improved the presentation of the content of Section 4.

---

> > ### Comment · Reviewer_nbQE · 2025-09-11
> > **Response**
> >
> > Many thanks for your response. I took a look at your response and believe that the paper can be published and do not have any further comment.

---

### Review · Reviewer_AFa3 · 2025-07-29

**Summary Of Contributions:**

The paper revisits the JL lemma-based dimensionality reduction and asks two intertwined questions:
How general can the distribution of the projection entries be while keeping the classical JL dimension bound? and how sparse can we make those projection matrices—without paying too high a price in target dimension?
Their main contribution is the new Chernoff‑style bound for the mean of squared sub‑Gaussian variables (Proposition 1), that underpins the proof and may be reused beyond this context. Extending the analysis, the authors characterise when a projection matrix with only a fraction q of non‑zeros preserves pairwise distances: essentially, the product qs (matrix sparsity times data sparsity) must be large enough. They provide matching upper (Theorems 2, 3) and lower (Theorem 4) bounds, showing the condition is “essentially not improvable.”

**Audience:**

Yes

**Broader Impact Concerns:**

Not applicable.

**Claims And Evidence:**

Yes

**Requested Changes:**

See above points (1) to (3), but these are mostly discussion points.

**Strengths And Weaknesses:**

In Prop 1, the proof is elementary yet tight to third order in \varepsilon^3; no Orlicz norms or Hanson-Wright are needed here. Then using a combinatorial uniion bound along with Prop 1, Theorem 1 is shown. Then Theorem 2 shows that quasi‑isometry holds as soon as q ⁣\geq ⁣max⁡ ∥xi​−xj​∥_\infty/∥xi​−xj​∥_2 times a poly-log(nd) and poly epsilon. This technique is neat and provides additional geometric insight about the (2,\infty)-norm used in high-dimensional statistics literature. Although these techniques are not new, the way they are organized in the current paper is indeed a nice technical piece to the community. I think the paper is quite well written, I only have a few comments/questions:

(1)The authors admit that upper deviations follow directly from Chernoff, whereas lower deviations need ad‑hoc manipulation, is it possible to provide some discussion on the limitation of this kind of manipualtion, specifically extensions to heavy‑tailed, dependent, or merely bounded‑moment distributions?
(2)The impossibility result is proved for a specific Bernoulli-Rademacher scheme; it is unclear whether other zero‑mean, sub‑Gaussian sparsification rules could circumvent the bound. WHen considering the equivalence of Bernoulli width and Gaussian widths (e.g., https://arxiv.org/abs/1305.4292), does it mean that the impossibility result also extends to other sub-Gaussian sparsifications?
(3)I like the connection to HW inequality which turns the JL error into a quadratic‑form deviation where nothing stronger can work because the lower‑bound in Theorem 4 shows the inequality reverses when q<1/s. Can the authors comment on whether/how this can be extended into matrix case (e.g., following J.Tropp's series of work extending the HW into matrix case)?

---

> ### Author Response · Authors · 2025-08-05
> **response**
>
> We thank all three reviewers for their positive feedback and constructive comments. We tried our best to improve the paper thanks to their numerous suggestions.
> *  Regarding extensions of the sparsity lower-bound to heavy‑tailed distributions:
> of course, heavier tails will yield less concentration and hence require larger target dimensions to ensure the same guarantees. With exponential moments (but no $1$-sub-Gaussianity), one obtains slower rate functions in the exponential deviations, and thus different (larger) variants of Condition (2). If you consider the case where we only assume that $\mathbb{E}[|A_{ik}|^2] = 1$ and that $\mathbb{E}[|A_{ik}|^u] \leq C$ for some constant $C$ and $u \geq 4$, the lower bound arguments for the non-sparse case remain valid. However, obtaining exponential tail bounds seems impossible in general. Nevertheless, these moment conditions imply $\mathbb{P}(|A_{ik}| > t) \leq Ct^{-u}$, which suggests an interesting research direction: investigating achievable bounds through truncation techniques for heavy tailed distributions. Specifically, one could define $\tilde{A}_{ik} = A_{ik}\mathbf{1}\{|A_{ik}| \leq b\}$ for some appropriately chosen threshold $b$. Other settings with e.g. dependencies would also be of interest, even if we are not sure to see which would be their impact in practice. On the other hand, sparsifying random matrices can be very helpful for reducing memory usage and computing random projections much faster.
>
> * Regarding the article \emph{On the boundedness of Bernoulli processes} that you mention: we were not aware of this paper, and it seems indeed interesting to investigate other sub-Gaussian sparsifications! We will consider this in future work -- a priori, we believe that it would be hard to circumvent this upper bound if no stronger assumption is made on the set of points $\{x_1, \ldots, x_n\}$.
>
> * Regarding an extension of HW to  the  matrix case : this is another very interesting perspective. We do not yet learn about matrix versions of the Johnson-Lindenstrauss lemma. Of course, if the objective is to project $a \times b$ matrices into a low-dimensional space, where the distance is measured by the Frobenius norm, then JL lemma and our results apply with $p = ab$... More interesting answers will necessitate more time from us!

---

### Decision · Action_Editor_juPf · 2025-09-23

**Recommendation:** Accept with minor revision

**Audience:**

Yes

**Audience Explanation:**

The paper offers a useful, streamlined treatment of JL under sub-Gaussianity and a careful analysis of sparsity trade-offs that will be of interest to both theory-minded readers and practitioners choosing projection schemes. I recommend publication, with the authors keeping the present clarifications and (optionally) adding a short numerical illustration figure in the camera-ready to make the sparsity–dimension trade-off more tangible (not required for acceptance).

**Claims And Evidence:**

Yes

**Claims Explanation:**

The paper gives a clean, elementary proof of the Johnson–Lindenstrauss (JL) lemma under sub-Gaussian assumptions and extends the analysis to sparse random projections. Beyond rederiving JL with simplified tools, the work makes explicit how the data’s own sparsity interacts with projection sparsity, clarifying when and how extremely sparse sketches remain reliable. The revised version also clarifies positioning relative to prior approaches.

The proof techniques are simple and pedagogically valuable. The authors’ revisions further improved the introduction and contribution list.
The paper delineates when sparse projections are viable and quantifies their dimensional overhead, which is practically useful for large-scale applications of random projections.

The weaknesses noted by reviewers are:
1) The results are primarily theoretical; empirical validation or numerical illustrations are minimal.
2) The reliance on sub-Gaussianity may limit immediate applicability to heavy-tailed settings; constants and tightness could be further discussed.
3) Earlier drafts needed clearer contribution framing and related-work comparisons; these have been substantially improved in the revision.

The authors have addressed the key reviewer requests by: (i) clarifying contributions and purpose in the introduction; (ii) improving Section 4’s presentation; (iii) adding comparison with state-of-the-art approaches; and (iv) specifying the primarily theoretical intent of Theorem 2. These changes materially strengthen the paper’s readability and contextualization without altering its core technical claims.